# TopoZero: Digging into Topology Alignment on Zero-Shot Learning

## Abstract

Common space learning, associating semantic and visual domains in a common latent space, is essential to transfer knowledge from seen classes to unseen ones on Zero-Shot Learning (ZSL) realm. Existing methods for common space learning rely heavily on structure alignment due to the heterogeneous nature between semantic and visual domains, but the existing design is sub-optimal. In this paper, we utilize persistent homology to investigate geometry structure alignment, and observe two following issues: (i) The sampled mini-batch data points present a distinct structure gap compared to global data points, thus the learned structure alignment space inevitably neglects abundant and accurate global structure information. (ii) The latent visual and semantic space fail to preserve multiple dimensional geometry structure, especially high dimensional structure information. To address the first issue, we propose a Topology-guided Sampling Strategy (TGSS) to mitigate the gap between sampled and global data points. Both theoretical analyses and empirical results guarantee the effectiveness of the TGSS. To solve the second issue, we introduce a Topology Alignment Module (TAM) to preserve multi-dimensional geometry structure in latent visual and semantic space, respectively. The proposed method is dubbed TopoZero. Empirically, our TopoZero achieves superior performance on three authoritative ZSL benchmark datasets.

## 1 Introduction

Given a large amount of training data, deep learning has exhibited excellent performance on various vision tasks, e.g., image recognition He et al. (2016); Dosovitskiy et al. (2020), object detection Lin et al. (2017); Liu et al. (2021), and instance segmentation He et al. (2017); Bolya et al. (2019). However, when considering a more realistic situation, e.g., the testing class does not appear at the training stage, the deep learning model fails to give a prediction on these novel classes. To remedy this, some pioneering researchers Lampert et al. (2014); Mikolov et al. (2013) point out that the auxiliary semantic information (sentence embeddings and attribute vectors) is available for both seen and unseen classes. Thus, by employing this common semantic representation, Zero-Shot Learning (ZSL) was proposed to transfer knowledge from seen classes to unseen ones.

Common space learning, enabling a significant alignment between semantic and visual information on the common embedding space, is a mainstream algorithm for ZSL. Existing approaches for common space learning can be divided into two categories: algorithms with 1) distribution alignment and 2) structure and distribution alignment. Typical methods in the first category employ various encoding networks to directly align the distribution between visual and semantic domains, e.g., variational autoencoder in Schönfeld et al. (2019), bidirectional latent embedding framework in Wang & Chen (2017), and deep visual-semantic embedding network in Tsai et al. (2017). Even though these methods encourage distribution alignment between visual and semantic domains, the alignment on the geometry structure is usually neglected. Note that the structure gap naturally exists in these two domains due to their heterogeneous nature Chen et al. (2021c). To mitigate the structure gap for promoting alignment between visual and semantic domains, HSVA Chen et al. (2021c) was proposed and become a pioneering work in the second category. Inspired by the successful structure alignment work Lee et al. (2019) in unsupervised domain adaptation, HSVA introduces a novel hierarchical semantic-visual adaptation framework to align the structure and distribution progressively.

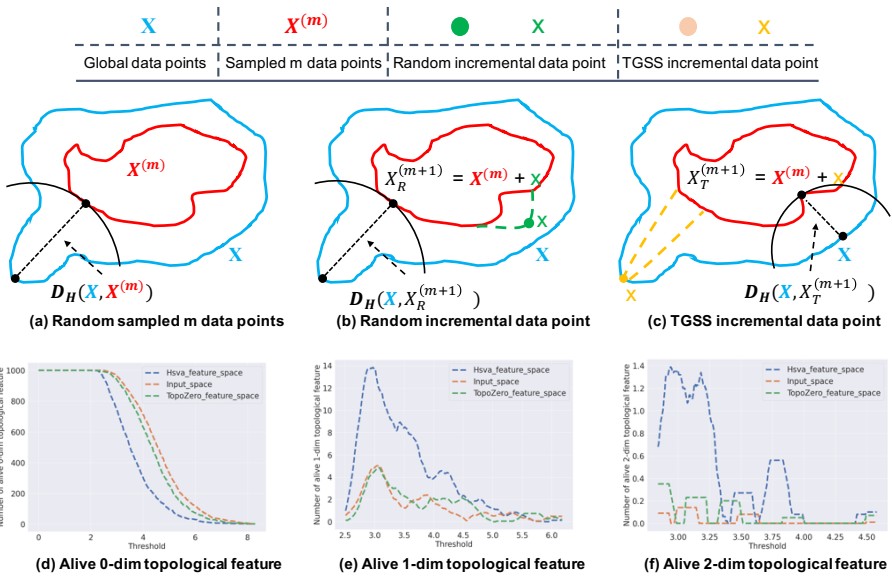

Figure 1: Motivation Illustration. (a)-(c) Based on the same random sampled data points $X^{(m)}$ in (a), the sampled batch data points from our Topological-guided Sampling Strategy (TGSS) are closer to the global data points compared to those sampled from random sampling strategy ($D_H(X, X_R^{(m+1)}) < D_H(X, X_T^{(m+1)})$). Combining this illustrator example with our theoretical analysis guarantees that our TGSS can mitigate the structure gap between mini-batch and global data points. (d)-(f) Compared to the input space, HSVA latent space can only preserve 0-dimensional topological features, indicating some high dimensional structure representation is lost during the dimension reduction phase. In contrast, our TopoZero latent space can preserve more accurate topological features by taking advantage of our proposed Topology Alignment Module.

Although HSVA empirically works well, we discover that there exist two issues in HSVA's structure alignment module. To clarify our findings clearly, we first introduce some background information in terms of Persistent Homology Zomorodian & Carlsson (2005). Persistent homology is a tool for computing topological features[1] of a data set at different spatial resolutions. More persistent features can be found over a wide range of spatial scales and represent true features of the underlying geometry space. We first introduce the concept of simplicial homology. For a simplicial complex $\mathcal{R}$, i.e. a generalised graph with higher-order connectivity information such as cliques, simplicial homology employs matrix reduction algorithms to assign $\mathcal{R}$ a family of groups, namely homology groups. The $d$-th homology group $\mathcal{H}_d(\mathcal{R})$ of $\mathcal{R}$ contains $d$-dimensional topological features, such as connected components ($d = 0$), cycles/tunnels ($d = 1$), and voids ($d = 2$). Homology groups are typically summarised by their ranks, thereby obtaining a simple invariant signature of a manifold. For example, a circle in $\mathbb{R}^2$ has one feature with $d = 1$ (a cycle), and one feature with $d = 0$ (a connected component). Based on these background knowledge, we further introduce how to compute a Persistent Homology when given a point cloud $X$. Firstly, we denote the Vietoris-Rips complex Vietoris (1927) of $X$ at scale $\epsilon$ as $\mathcal{V}_\epsilon(X)$. Then, we can obtain the Persistent Homology $\text{PH}(\mathcal{V}_\epsilon(X))$ of a Vietoris-Rips complex $\mathcal{V}_\epsilon(X)$, which consists of persistence diagrams $\{\mathcal{D}_1, \mathcal{D}_2, ...\}$ and persistence pairs $\{\pi_1, \pi_2, ...\}$. The $d$-dimensional persistence diagram $\mathcal{D}_d$ contains coordinates with the form $(a, b)$, where $a$ refers to a threshold $\epsilon$ at which a $d$-dimensional topological feature appears and $b$ refers to a threshold $\epsilon'$ at which it disappears. The d-dimensional persistence pairs contains indices $(i, j)$ corresponding to simplices $s_i, s_j \in \mathcal{V}_\epsilon(X)$, which create and destroy the corresponding topological features determined by $(a, b) \in \mathcal{D}_d$. Note that more detailed background knowledge (e.g., simplex, Vietoris-Rips complex) is introduced in Section A.

---

[1] Connectivity-based features, e.g., connected components in 0-dimensional, cycles in 1-dimensional, and voids in 2-dimensional topological features

Based on the powerful geometry feature analysis ability of Persistent Homology, we discover 2 problems in the existing state-of-the-art (sota) structure alignment module Chen et al. (2021c): (i) Due to the limitation of batch size, the underlying geometry structure of mini-batch samples can not represent global samples'. Thus, when applying structure alignment metric (i.e., sliced Wasserstein discrepancy Lee et al. (2019)) on random sampled [2] mini-batch visual and semantic data points, we can only achieve a local-level structure alignment, indicating the accurate global geometry information is lost inevitably. (ii) HSVA utilizes sliced Wasserstein discrepancy to align latent visual and semantic space for bridging structure alignment. Actually, this implementation requires an assumption that the latent visual and semantic space can represent their underlying geometry structure adequately. To verify the correctness of this assumption, we adopt persistent homology to visualize the underlying geometry structure of input space and latent space on the visual domain. As shown in Fig. 1 (d) - (f), there is a distinct gap between the blue dash line and the orange dash line, which is further expanded in the latter two images, representing that the HSVA latent visual space loses abundant geometry structure, especially for 1-dimensional and 2-dimensional topological features. The rationale is that after dimensionality reduction (namely curse of dimensionality Wang & Chen (2017) ), the topological structure is difficult to maintain.

In this paper, we devise a TopoZero framework to achieve a more desirable structure alignment by solving 2 aforementioned issues. Concretely, our TopoZero adopts CADA-VAE Schönfeld et al. (2019) as the distribution alignment module and develops a Topology Alignment Module (TAM) with 2 following novelties. (i) To alleviate the structure gap between the sampled mini-batch data points and global data points, we propose a Topology-guided Sampling Strategy (TGSS) to explicitly and progressively mine the topology-preserving data point into the sampled mini-batch data point. Moreover, the theoretical analysis illustrated in Section A guarantees the advantage of our TGSS. Besides, as shown in Fig. 1 (b) - (c), we further visualize the advantage of our TGSS in an illustrator example: based on the same random sampled data points $X^{(m)}$, $X_T^{(m+1)}$ and $X_R^{(m+1)}$ are constructed by our TGSS and random sampling strategy, respectively. Obviously, the Hausdorff Distance [3] $D_H(X, X_T^{(m+1)})$ between $X_T^{(m+1)}$ and global data points $X$ is bounded by $D_H(X, X_R^{(m+1)})$, indicating our TGSS can alleviate the gap between sampled data points and global data points compared to random sampling strategy. (ii) To preserve the topological structure for visual and semantic latent space, we develop a dual topological-aware branch as well as a topological-preserving loss to learn a topological-invariant latent representation. Moreover, based on the open-source tool Ripser [4], we compute the persistent homology to analyze the multi-dimensional topological features from input space, HSVA latent structure space, and TopoZero latent structure space on the visual domain. Given a set of data points, ripser can compute the corresponding persistent homology, which consists of persistence diagrams $\{\pi_1, \pi_2, ...\}$ and persistence pairs $\{D_1, D_2, ...\}$. Thus based on the obtained persistence diagrams and persistence pairs, we can calculate the number of alive 0/1/2-dimensional topological features under different threshold $\epsilon$. As such, we draw the Fig. ·1 (d)-(f), where the line represents the trend of the number of alive topological features under different threshold $\epsilon$. As revealed from these visualization results, by taking advantage of our proposed TAM, the multi-dimensional topology feature gap between our TopoZero latent space and input space is negligible.

## 2 RELATED WORKS

**Zero-Shot Learning.** In recent years, the ZSL realm has attracted many researchers' attention Zhang & Saligrama (2016); Li et al. (2017); Zhu et al. (2019a); Fu et al. (2015); Ye & Guo (2017); Yu & Lee (2019b); Chen et al. (2018). One typical branch to solve the ZSL problem is learning a common embedding space for aligning semantic and visual domains, termed common space learning. Early common space learning methods focus on framework designation for better distribution alignment. Wang *et al.* Wang & Chen (2017) have proposed a bidirectional latent embedding framework with two subsequent learning stages. Liu et al. (2018) maps visual features and semantic representations of class prototypes into a common embedding space to guarantee the seen data is compatible with seen and unseen classes. CADA-VAE Liu et al. (2018) have demonstrated that

---

[2] Existing methods all adopt random sampling strategy to generate mini-batch data points.

[3] A metric that can measure the bounded distance between two persistence diagrams.

[4] Available at `https://github.com/Ripser/ripser`.

only two variational autoencoders as well as a distribution alignment loss, can achieve a significant distribution alignment in a common space. However, as pointed out from HSVA Chen et al. (2021c), due to the heterogeneous nature of the feature representations in semantic and visual domains, the distribution and structure variation intrinsically exists. Motivated by this, Chen *et al*. Chen et al. (2021c) propose a hierarchical semantic-visual adaptation framework for aligning structure and distribution progressively. Thus, the structure alignment in ZSL emerges with a new state-of-the-art performance on the task of common space learning.

**Persistent Homology.** Persistent homology, a tool for topological data analysis, is used for understanding topological features at different dimension. Concretely, persistent homology can detect multi-dimensional topological features (holes, circles, connected components) under various dimensions for the underlying manifold of a set of sampled data points. Based on this property, persistent homology has been applied to a vast body of scenarios, e.g., characterizing graphs in Archambault et al. (2007); Carrière et al. (2020); Li et al. (2012), analysing underlying manifolds in Bae et al. (2017); Futagami et al. (2019), topological preserving autoencoder in Moor et al. (2020). In this paper, by leveraging persistent homology, we discover that the latent visual and semantic space can not preserve multi-dimensional topological features. Furthermore, to improve the geometry representation of latent space in both domains, we propose a Topology Alignment Module for encoding multi-dimensional topological representation explicitly.

## 3 METHODOLOGY

To begin with, we formulate the task of ZSL. Assume we have a set of seen samples $S$ for training, and a set of unseen samples $U$ for testing only, where $S = \{(x^s, y^s, a^s) \mid x^s \in X^s, y^s \in Y^s, a^s \in \mathcal{A}\}$ be a training set. $x^s$ is seen image feature, which is extracted from the pre-trained CNN backbone (ResNet-101 He et al. (2016) is adopted in this paper). $y^s$ and $a^s$ are $x^s$ corresponding class label and semantic vector, respectively. Analogously, let $U = \{(x^u, y^u) \mid x^u \in X^u, y^u \in Y^u\}$. Note that $Y^s \cap Y^u = \emptyset$. The objectiveness of conventional ZSL (CZSL) is to learn a classifier for mapping unseen image features into unseen categories, i.e., $\mathcal{F}_{czsl} : \mathcal{X}^u \to \mathcal{Y}^u$, while the challenging generalized ZSL (GZSL) focus on learning a classifier to map image features to both seen and unseen categories, i.e., $\mathcal{F}_{gzsl} : \mathcal{X} \to \mathcal{Y}^u \cup \mathcal{Y}^s$.

As shown in Fig. 2, our TopoZero contains two parallel alignment modules, Distribution Alignment Module and Topology Alignment Module Specifically, we directly adopt the architecture of CADA-VAE Schönfeld et al. (2019) as our Distribution Alignment Module. While for our TAD, topology-guided sampling strategy and dual topological-aware branch are proposed to mitigate the geometry structure gap between mini-batch and global data points and preserve multi-dimensional topological structure on both visual and semantic domains, respectively.

### 3.1 TOPOLOGY-GUIDED SAMPLING STRATEGY

To bridge a structure gap between mini-batch and global data points, we propose a Topology-guided Sampling Strategy (TGSS) as well as a theoretical analysis to guarantee its superiority.

#### 3.1.1 DESCRIPTION

Algorithm 1 describes how our TGSS samples mini-batch samples from global data points. First, we random sample $b/2$ [5] data points ($X_{b2}$) from global training samples ($X$). After that, we select the incremental data point $x_{max}$ according to Equ. 1. Then, we construct a set of candidate set ($\mathcal{C}$) by Equ. 2 and random sample $b/2 - 1$ data points from $\mathcal{C}$ to form $\mathcal{C}_{mini}$. Finally, the mini-batch sampled data points are constructed by integrating $X_{b2}$, $x_{max}$ and $\mathcal{C}_{mini}$. The advantage of our TGSS relies heavily on the selection of $x_{max}$, which is proved by the following theoretical analysis.

$$\exists \, x_{max} \in X, x'_{max} \in X_{b2}, s.t. \ dist(x_{max}, x'_{max}) = \mathrm{d_H}(X, X_{b2}) \tag{1}$$

$$\mathcal{C}(x_{max}, d) = \{\{x_0, ..., x_k\}, x_i \in X, x_i \notin \mathcal{T} \ || \ dist(x_i, x_{max}) < d\} \tag{2}$$

---

[5]$b$ represents the size of batch training samples

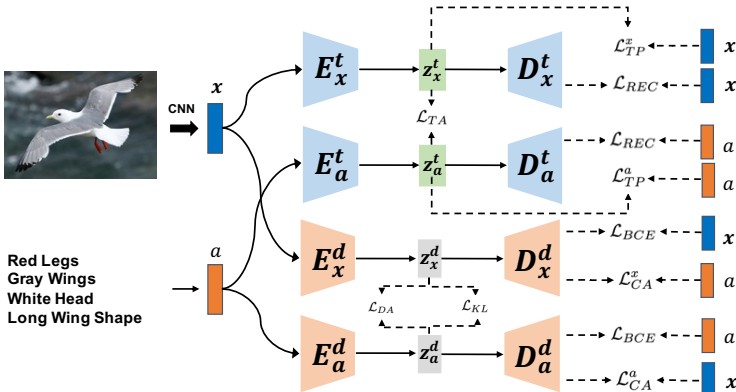

Figure 2: The proposed TopoZero framework. Based on our proposed TGSS sampling strategy, we can obtain a batch of visual features $x$ and corresponding semantic embeddings $a$. Then $x$ and $a$ are fed into the parallel topology alignment module and distribution alignment module. For TAD, the encoder $E_x^t$ and $E_a^t$ first encode $x$ and $a$ to get latent topological visual representation $z_x^t$ and topological semantic representation $z_a^t$, respectively. Then the decoder $D_x^t$ and $D_a^t$ decode $z_x^t$ and $z_a^t$ to reconstruct visual and semantic feature, which is optimized by reconstuction loss $\mathcal{L}_{AE}^a$ and $\mathcal{L}_{AE}^x$. The visual and semantic latent topological representation is optimized by $\mathcal{L}_{TP}^x$ and $\mathcal{L}_{TP}^a$ to preserve multi-dimensional structure information. $\mathcal{L}_{TA}$ is also applied to align $z_x^t$ and $z_a^t$. For the distribution alignment module, we adopt the framework of CADA-VAE, which consists of two variational autoencoders and optimized by $\mathcal{L}_{BCE}$, $\mathcal{L}_{KL}$, $\mathcal{L}_{DA}$, and $\mathcal{L}_{CA}$.

where $\mathcal{T}$ denotes a set of sampled data points from $X$ and $dist$ represents distance metric (Euclidean Distance in this paer). $d_H$ refers to the Hausdorff distance Huttenlocher et al. (1993) between $X$ and $X_{b2}$. Then, we revisit the definition of Hausdorff Distance that $d_H(X, Y) = \max \left\{ \sup_{x \in X} d(x, Y), \sup_{y \in Y} d(X, y) \right\}$, which measures how far two subsets of a metric space are from each other. Informally speaking, the $x_{max}$ represents the farthest data point in $X$ to the sampled $X_{b2}$ when adopting Hausdorff Distance metric. Thus, by integrating the $x_{max}$ into $X_{b2}$, the Hausdorff Distance between $X$ and $X_{b2}$ can be reduced, indicating the gap between sampled and global data points is also mitigated according to Theorem 1. Moreover, considering that the advantage of our TGSS relies heavily on the selection of $x_{max}$, we provide a theoretical analysis to guarantee its superiority. Besides, the introduction of $\mathcal{C}_{mini}$ is to maintain the representation of local topology structure surrounding from $x_{max}$.

### 3.1.2 THEORETICAL ANALYSIS FOR TGSS

The core design of our TGSS is the procedure of selection $x_{max}$ (line 4 in Algorithm 1), which can eliminate the structure gap compared to random sampling strategy. Here, we further provide a theoretical analysis to guarantee the advantage of this selection procedure. Before we carry out our analysis, we define a few important definitions and notations. For a point cloud $X := \{x_1, \ldots, x_m\} \subseteq R^d$, denote $X^{(m)}$ be a subsample of $X$ with cardinality m. Based on $X^{(m)}$ and the procedure of TGSS's selection $x_{max}$, the constructed set is denoted as $X_T^{(m+1)}$. While for random sampling strategy, we have $X_R^{(m+1)}$. Thus, we have:

$$X_T^{(m+1)} = \{X^{(m)} \cup x, x = x_{max}\} \tag{3}$$

$$X_R^{(m+1)} = \{X^{(m)} \cup x, x \in X \backslash X^{(m)}\} \tag{4}$$

where $x_{max}$ is defined in Equ. 1

**Theorem 1.** *Moor et al. (2020). Let $X$ be a point cloud of cardinality n and $X^{(m)}$ be one subsample of $X$ of cardinality m, i.e. $X^{(m)} \subseteq X$, sampled without replacement. We can bound the probability of the persistence diagrams of $X^{(m)}$ exceeding a threshold in terms of the bottleneck distance as*

$$P(d_b(\mathcal{D}^X, \mathcal{D}^{X^{(m)}}) > \epsilon) \leq P(d_H(X, X^{(m)}) > 2\epsilon) \tag{5}$$

---

**Algorithm 1** Topology-guided Sampling Strategy

---

**Input:**
    $X$ is a set of whole training samples.
    $b$ is the size of batch training samples.
**Output:**
    $\mathcal{T}$ is prepared mini-batch samples for an epoch.

1:  init $\mathcal{T}$: $\mathcal{T} \leftarrow \varnothing$
2:  **for** each iteration in epoch **do**
3:     random sample $b\,/\,2$ training data from $X \backslash \mathcal{T}$: $X_{b2}$;
4:     compute the incremental data point $x_{max}$ by Equ. 1;
5:     construct a set of data points by Equ. 2: $\mathcal{C} = \mathcal{C}(x_{max}, d_H(X_{b2}, X))$;
6:     $X_{b2} = X_{b2} \cup x_{max}$;
7:     **if** $len(\mathcal{C}) < b/2 - 1$ **then**
8:        $\mathcal{M} \leftarrow$ random select $b/2 - 1 - len(\mathcal{C})$ data points from $X/X_{b2}$;
9:        $X_{b2} = X_{b2} \cup \mathcal{C} \cup \mathcal{M}$;
10:    **else**
11:       $\mathcal{M} \leftarrow$ random select $b/2 - 1$ data points from $\mathcal{C}$;
12:       $X_{b2} = X_{b2} \cup \mathcal{M}$;
13:    **end if**
14:    $\mathcal{T} = \mathcal{T} \cup X_{b2}$
15:  **end for**
16:  **return** $\mathcal{T}$;

---

**Theorem 2.** *Let* $\mathbf{A}_{X, X_T^{(m+1)}} \in R^{n \times (m+1)}$ *be the distance matrix between samples of* $X$ *and* $X_T^{(m+1)}$, *and* $\mathbf{A}_{X, X_R^{(m+1)}} \in R^{n \times (m+1)}$ *be the distance matrix between samples of* $X$ *and* $X_R^{(m+1)}$. *The* $X_T^{(m+1)}$ *and* $X_R^{(m+1)}$ *are both sorted to ensure that the first (m+1) rows correspond to the columns of the m subsampled points with diagonal elements* $a_{ii} = 0$. *Assume that the entries* $a_{ij}$ *in both matrix are independent and follow a same distance distribution* $F_D$ *when* $i > (m+1)$. *For* $\mathbf{A}_{X, X_T^{(m+1)}}$, *the minimal distances* $\delta_i'$ *for rows with* $i > (m+1)$ *follow a distribution* $F_{\Delta'}$. *Letting* $Z' := \max_{1 \leq i \leq n} \delta_i'$ *with a corresponding distribution* $F_Z'$. *For* $\mathbf{A}_{X, X_R^{(m+1)}}$, *the minimal distances* $\delta_i''$ *for rows with* $i > (m+1)$ *follow a distribution* $F_{\Delta''}$. *Letting* $Z'' := \max_{1 \leq i \leq n} \delta_i''$ *with a corresponding distribution* $F_Z''$, *the expected Hausdorff distance between* $X$ *and* $X_T^{(m+1)}$ *is bounded by:*

$$\mathrm{E}[d_H(X, X_T^{(m+1)})] \leq \mathrm{E}[d_H(X, X_R^{(m+1)})] \tag{6}$$

We include its proof in Section A. Theorem. 2 illustrates that compared to random sampling strategy $(X_R^{(m+1)})$, the sampled batch data points $(X_T^{(m+1)})$ from our TGSS are closer to the global data points $X$ with Hausdorff Distance metric, which constitutes the upper bound of bottleneck distance between two persistence diagrams (Theorem 1). Thus, since bottleneck distance is usually used to measure the distance between two persistence diagrams in the topological space Beketayev et al. (2014); Bubenik et al. (2010), we can conclude that compared to random sampling strategy, the sampled batch data points from our TGSS are closer to the global data points in the topological space.

## 3.2 Topology Alignment Module

As shown in Fig. 1 (a)-(c), HSVA, a state-of-the-art common space learning method by taking structure alignment into account, fails to preserve multi-dimensional topological features. Specifically, the terrible structure representation in the latent space inevitably leads to a sub-optimal structure alignment. To remedy this, we propose a Topology Alignment Module, consisting of a dual topology-aware branch and a topology-preserving loss, to encode multi-dimensional topological information into latent visual and semantic space for conducting a more desirable structure alignment.

Our Dual Topology-aware Branch is illustrated in Fig. 2, which contains two autoencoders for obtaining topological-aware latent representation in visual and semantic domains. Specifically, the encoder $E_x^t$ / $E_a^t$ encodes image feature (x) / semantic vector (a) into latent space and obtain visual

and semantic topological-aware latent representation $Z_a^{(m)}$ and $Z_v^{(m)}$. After that, the decoder $D_x^t$ / $D_a^t$ decodes $Z_a^{(m)}$ / $Z_v^{(m)}$ for reconstructing the latent representation into $x$ / $a$. We first apply reconstruction loss to optimize our Dual Topology-aware Branch:

$$\mathcal{L}_{AE}^x = \mathcal{L}_{REC} = \|D_x^t(E_x^t(x)) - x\|^2 \tag{7}$$

$$\mathcal{L}_{AE}^a = \mathcal{L}_{REC} = \|D_a^t(E_a^t(a)) - a\|^2 \tag{8}$$

Then we utilize the topology-preserving loss proposed by Moor et al. (2020) to preserve multiple dimensional topological features on the latent visual and semantic space, which is calculated by the following steps: 1) Given a batch of visual feature $X_v^{(m)}$ and semantic embeddings $X_a^{(m)}$, our dual topology-aware branch can obtain corresponding latent representation, $Z_v^{(m)}$ and $Z_a^{(m)}$; 2) We calculate the distance matrix between samples of $X_v^{(m)}$ and $X_v^{(m)}$, termed $A_{X_v^{(m)}}$. The corresponding persistent homology of $X_v^{(m)}$ is recorded as $\text{PH}(\mathcal{V}_\epsilon(X_v^{(m)})) = (\mathcal{D}^{X_v^{(m)}}, \pi^{X_v^{(m)}})$. Analogously, for $X_a^{(m)}$, $Z_v^{(m)}$ and $Z_a^{(m)}$, we can obtain corresponding distance matrix $A_{X_a^{(m)}}$, $A_{Z_v^{(m)}}$ and $A_{Z_a^{(m)}}$, persistence pairings $\pi^{X_a^{(m)}}$, $\pi^{Z_v^{(m)}}$ and $\pi^{Z_a^{(m)}}$; 3) Finally, we retrieve the value of 0-dimensional / 1-dimensional / 2-dimensional persistence diagram [6] from distance matrix with indices provided by the persistence pairings, namely $\mathcal{D}_0^{X_v^{(m)}} \simeq \mathbf{A}^{X_v^{(m)}}[\pi_0^{X_v^{(m)}}]$. Through this computation process, we get the 0/1/2 -dimensional persistence diagrams in $X_v^{(m)}$ $X_a^{(m)}$, $Z_v^{(m)}$ and $Z_a^{(m)}$, which are optimized by the following topology-preserving loss:

$$\mathcal{L}_{TP}^x = \sum_{i=0}^2 \| \mathcal{D}_i^{X_v^{(m)}} - \mathcal{D}_i^{Z_v^{(m)}} \|^2, \tag{9}$$

$$\mathcal{L}_{TP}^a = \sum_{i=0}^2 \| \mathcal{D}_i^{X_a^{(m)}} - \mathcal{D}_i^{Z_a^{(m)}} \|^2 \tag{10}$$

Finally, to encourage interaction between visual and semantic domains in the topological space, we directly minimize the L2 distance between latent visual topological representation and latent semantic topological representation:

$$\mathcal{L}_{TA} = \|Z_v^{(m)} - Z_a^{(m)}\|^2 \tag{11}$$

$$\tag{12}$$

## 3.3 Distribution Alignment Module

Since CADA-VAE Schonfeld et al. (2019) serves as our distribution alignment module, we directly revisit it in our framework. Our distribution alignment module adopts two variational autoencoders Kingma & Welling (2014) to obtain latent representation in visual and semantic domains, respectively. Concretely, the encoder $E_x^d$ / $E_a^d$ encodes image feature (x) / semantic vector (a) into latent space and obtain visual and semantic latent representation $z_x^d$ and $z_a^d$. Then, the decoder $D_x^d$ / $D_a^d$ decodes $z_x^d$ / $z_a^d$ for reconstructing the latent representation into $x$ / $a$. We apply standard VAE loss to optimize:

$$\mathcal{L}_{VAE}^x = \mathcal{L}_{BCE} - \beta\mathcal{L}_{KL} = \mathbb{E}_{E_x^d(x)}[\log D_x^d(z_x^d)] - \beta D_{KL}(E_x^d(x)\|p(z)) \tag{13}$$

$$\mathcal{L}_{VAE}^a = \mathcal{L}_{BCE} - \beta\mathcal{L}_{KL} = \mathbb{E}_{E_a^d(a)}[\log D_a^d(z_a^d)] - \beta D_{KL}(E_a^d(a)\|p(z)) \tag{14}$$

where $D_{KL}$ represents the Kullback-Leibler divergence and $p(z)$ is a prior distribution (standard Gaussian distribution $\mathcal{N}(0, 1)$ in this paper). The binary cross-entropy loss $\mathcal{L}_{BCE}$ is served as the reconstruction loss. Following Schonfeld et al. (2019), $\beta$ serves as the balanced weight to measure the importance of $D_{KL}$.

Distribution alignment loss is formulated as:

$$\mathcal{L}_{DA} = \left( \|\mu^x - \mu^a\|_2^2 + \left\| (\delta^x)^{\frac{1}{2}} - (\delta^a)^{\frac{1}{2}} \right\|_F^2 \right)^{\frac{1}{2}} \tag{15}$$

---

[6]Due to the page limited, we provide a more detailed computation process in Section A

where $\| \cdot \|_F^2$ is the squared matrix Frobenius norm, and cross-alignment loss is formulated as:

$$\mathcal{L}_{CA}^x = |x - D_x^d(E_a^d(a))| \tag{16}$$

$$\mathcal{L}_{CA}^a = |a - D_a^d(E_x^d(x))| \tag{17}$$

## 3.4 TOPOZERO OBJECTIVE FUNCTION

Our TopoZero is optimized by the following objective function:

$$
\begin{aligned}
\mathcal{L}_{TopoZero} = {} & \mathcal{L}_{AE}^x + \mathcal{L}_{AE}^a + \lambda_1 * (\mathcal{L}_{TP}^x + \mathcal{L}_{TP}^a) + \lambda_2 * \mathcal{L}_{TA} \\
& + \lambda_3 * (\mathcal{L}_{CA}^x + \mathcal{L}_{CA}^a + L_{VAE}^x + \mathcal{L}_{VAE}^a + \mathcal{L}_{DA})
\end{aligned}
\tag{18}
$$

where $\lambda_1$, $\lambda_2$, and $\lambda_3$ are the balanced weight to measure the importance of each module in our TopoZero. In the branch of TAM, $\mathcal{L}_{AE}^x$ and $\mathcal{L}_{AE}^a$ aim to obtain the latent visual and semantic representation. $\mathcal{L}_{TP}^x$ and $\mathcal{L}_{TP}^a$ assist the latent visual and semantic representation to preserve multi-dimensional topology structure. $\mathcal{L}_{TA}$ associates semantic and visual latent representation in a common space. While for the branch of distribution alignment module, all the objective functions keep the same with those in CADA-VAE Zhu et al. (2019a).

## 3.5 ZERO-SHOT PREDICTION

After the optimization of TopoZero, we need to train $\mathcal{F}_{gzsl}$ and $\mathcal{F}_{czsl}$ for predicting unseen or seen samples. Given a seen image features $x^s$, we can obtain the latent distribution representation $z_{x^s}^d = E_x^d(x^s)$ with reparametrization trick Kingma & Welling (2014) and topological representation $z_{x^s}^t = E_x^t(x^s)$. Analoguously, for unseen image semantic vector $a^u$, we have $z_{a^u}^d$ and $z_{a^u}^t$. Then we concatenate $z_{x^s}^d$ and $z_{x^s}^t$ ($[z_{x^s}^d, z_{x^s}^t]$) to serve as seen training data and ($[z_{a^u}^d, z_{a^u}^t]$) for unseen one. After training $\mathcal{F}_{gzsl}$ and $\mathcal{F}_{czsl}$, we use $[z_{x^s}^d, z_{x^s}^t]$ and $[z_{x^u}^d, z_{x^u}^t]$ to inference.

## 4 EXPERIMENTS

In this section, we first elaborate on implementation details and 3 authoritative benchmark datasets in the field of ZSL. Then we compare our TopoZero with existing state-of-the-art ZSL methods. Finally, we provide some qualitative and quantitative analysis to illustrate the advantage of our TopoZero. Due to the limitation of page size, several parts are placed on Appendix A.

## 4.1 DATASETS AND IMPLEMENTATION

**Datasets.** We verify our TopoZero on 3 popular ZSL benchmark datasets, including CUB Welinder et al. (2010), SUN Patterson & Hays (2012), and AWA2 Xian et al. (2018a). CUB contains 11788 images of 200 bird classes (seen/unseen classes = 150/50) with 312 attributes. SUN consists of 14340 images from 717 classes (seen/unseen classes = 645/72) with 102 attributes. AWA2 includes 37322 images of 50 animal classes (seen/unseen classes = 40/10) with 85 attributes. Finally, we adopt the "split version 2.0" mode Xian et al. (2018b) to conduct data splits on CUB, SUN, and AWA2.

**Network Architecture.** As illustrated in Fig. 2, our TopoZero contains 2 Encoders and 2 Decoders, which are basic Multi-Layer Perceptions with 2 fully connected (FC) layers and 4096 hidden units. The dimension of latent variable in the distribution alignment and topology alignment module are both set 64. The architecture of CZSL and GZEL classifier is a single FC layer.

**Optimization Details.** Our TopoZero is optimized by Adam optimizer with an initial learning rate $10^{-4}$. The total training epoch of TopoZero is set 100 with a batch size 50. For training final CZSL and GZSL classifiers, the training epoch, batch size, and initial learning rate are set 25, 28, $10^{-3}$ respectively.

**Evaluation Protocols.** Following the standard evaluation protocol Xian et al. (2018a), our TopoZero is evaluated by the top-1 accuracy. For CZSL, we only compute the accuracy on unseen classes. While for GZSL, we both calculate the accuracy of seen and unseen classes.

Table 1: Results (%) of the state-of-the-art models on CUB, SUN and, AWA2 datasets. The best result is masked in **bold**. The symbol "–" indicates no available result.

| | **CUB** | | | | **SUN** | | | | **AWA2** | | | |
|---|---|---|---|---|---|---|---|---|---|---|---|---|
| | CZSL | GZSL | | | CZSL | GZSL | | | CZSL | GZSL | | |
| Methods | acc | U | S | H | acc | U | S | H | acc | U | S | H |
| **Non Common Space** | | | | | | | | | | | | |
| QFSL Song et al. (2018) | 58.8 | 33.3 | 48.1 | 39.4 | 56.2 | 30.9 | 18.5 | 23.1 | 63.5 | 52.1 | 72.8 | 60.7 |
| LDF Li et al. (2018) | 67.5 | 26.4 | 81.6 | 39.9 | – | – | – | – | 65.5 | 9.8 | 87.4 | 17.6 |
| SGMA Zhu et al. (2019b) | 71.0 | 36.7 | 71.3 | 48.5 | – | – | – | – | 68.8 | 37.6 | 87.1 | 52.5 |
| AREN Xie et al. (2019) | 71.8 | 38.9 | 78.7 | 52.1 | 60.6 | 19.0 | 38.8 | 25.5 | 67.9 | 15.6 | **92.9** | 26.7 |
| LFGAA Liu et al. (2019) | 67.6 | 36.2 | **80.9** | 50.0 | 61.5 | 18.5 | 40.0 | 25.3 | 68.1 | 27.0 | 93.4 | 41.9 |
| SP-AEN Chen et al. (2018) | 55.4 | 34.7 | 70.6 | 46.6 | 59.2 | 24.9 | 38.6 | 30.3 | 58.5 | 23.3 | 90.9 | 37.1 |
| PQZSL Li et al. (2019) | – | 43.2 | 51.4 | 46.9 | – | 35.1 | 35.3 | 35.2 | – | 31.7 | 70.9 | 43.8 |
| CRNet Zhang & Shi (2019) | – | 45.4 | 56.8 | 50.5 | – | 34.1 | 36.5 | 35.3 | – | – | – | – |
| IIR Cacheux et al. (2019) | 63.8 | 30.4 | 65.8 | 41.2 | 63.5 | 22.0 | 34.1 | 26.7 | 67.9 | 17.6 | 87.0 | 28.9 |
| DVBE Min et al. (2020) | – | 53.2 | 60.2 | 56.5 | – | 45.0 | 37.2 | 40.7 | – | 63.6 | 70.8 | 67.0 |
| FREE Chen et al. (2021b) | – | 55.7 | 59.9 | 57.7 | – | 47.4 | 37.2 | 41.7 | – | 60.4 | 75.4 | 67.1 |
| *Common Space* | | | | | | | | | | | | |
| DeViSE Frome et al. (2013) | - | 23.8 | 53.0 | 32.8 | - | 16.9 | 27.4 | 20.9 | - | 17.1 | 74.7 | 27.8 |
| ReViSE Tsai et al. (2017) | - | 37.6 | 28.3 | 32.3 | - | 24.3 | 20.1 | 22.0 | - | 46.4 | 39.7 | 42.8 |
| DCN Liu et al. (2018) | - | 28.4 | 60.7 | 38.7 | - | 25.5 | 37.0 | 30.2 | - | - | - | - |
| SGAL Yu & Lee (2019a) | - | 44.7 | 47.1 | 45.9 | - | 31.2 | **42.9** | 36.1 | - | **81.2** | 55.1 | 65.6 |
| CADA-VAE Schönfeld et al. (2019) | 57.9 | 51.6 | 53.5 | 52.4 | 61.6 | 47.2 | 35.7 | 40.6 | 62.6 | 51.6 | 53.5 | 52.4 |
| DOE-ZEL Geng et al. (2022) | - | - | - | - | - | - | - | - | 66.4 | - | - | 57.6 |
| VGSE Xu et al. (2022) | 56.8 | 24.1 | 45.7 | 31.5 | 41.1 | 25.5 | 35.7 | 29.8 | 66.7 | 45.7 | 66.7 | 54.2 |
| HSVA Chen et al. (2021c) | 62.8 | 52.7 | 58.3 | 55.3 | 63.8 | 48.6 | 39.0 | 43.3 | 70.6 | 59.3 | 76.6 | 66.8 |
| **TopoZero (Ours)** | **64.3** | **54.9** | 59.9 | **57.3** | **64.7** | **49.4** | 40.9 | **44.7** | **70.6** | 59.1 | 80.0 | **68.0** |

For determining the performance of GZSL in a unified criterion, the harmonic mean (defined as $H = (2 \times S \times U)/(S + U)$) is adopted in this paper.

### 4.2 COMPARISON WITH STATE-OF-THE-ARTS.

**Results on Conventional Zero-Shot Learning.** Tab. 1 reports the CZSL results of our TopoZero and recent state-of-the-art (sota) methods on 3 ZSL datasets. Considering that attribute-based sota methods Huynh & Elhamifar (2020); Chen et al. (2021a) exploit the advantage of pre-trained NLP models GloVE and generation-based sota methods Xian et al. (2018b); Yu et al. (2020) take advantage of data augmentation, methods involving these 2 branches are not taken into account in this part. Compared to methods only with distribution alignment, our TopoZero illustrates a significant improvement of 6.4%, 3.1%, and 8.0% on CUB, SUN, and AWA2 datasets at least. While compared to HSVA Chen et al. (2021c) with distribution and structure alignment, our TopoZero also achieves a great improvement of 1.5%, 0.9% on CUB and SUN datasets, respectively. Such a significant performance directly verifies the effectiveness of topology alignment for the ZSL task.

**Results on Generalized Zero-Shot Learning.** By looking at the challenging GZSL results in Tab. 2, our TopoZero also achieves a dominant harmonic mean performance of 57.3%, 44.7%, and 68.0% on CUB, SUN, and AWA2 datasets, respectively. Both superiority results of TopoZero on CZSL and GZSL settings demonstrate that our TopoZero is better than HSVA on structure alignment.

## 5 CONCLUSION

In this paper, we propose a TopoZero framework to improve structure alignment for common space learning methods. To begin with, we discover that existing structure alignment approaches confront two challenging issues: 1) sampled mini-batch data points present a distinct gap compared to global ones; 2) latent visual and semantic space lose some high-dimensional structure information due to the 'curse of dimensionality.' To solve these two problems, Topology-guided sampling strategy and Topology Alignment Module are proposed to construct our TopoZero. Furthermore, we provide a theoretical analysis as well as visualization results to guarantee the advantage of our TopoZero, namely excellent multi-dimensional topology-preserving and topology-alignment ability. Finally, The extensive and superior experiment results demonstrate that our TopoZero has a great potential to advance the ZSL community.

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

## A   APPENDIX

## B   PROOF OF THEOREM 2

*Proof.* First, we derive the distribution $F_{\Delta'}(y)$ and $F_{\Delta''}(y)$:

$$F_{\Delta'}(y) = \mathrm{P}(\delta_i' \leq y) = 1 - \mathrm{P}(\delta_i' > y) = 1 - \mathrm{P}(\min_{1 \leq j \leq m+1} a_{ij} > y) \tag{19}$$

$$= 1 - \mathrm{P}(\bigcap_j a_{ij} > y) = 1 - (1 - F_D(y))^{m+1} \tag{20}$$

$$= \begin{cases} 1 - (1 - F_D(y))^{m+1} & , y < \mathrm{E}[d_H(X, X^{(m)})] \\ 1 & , else \end{cases} \tag{21}$$

Analogously, we have:

$$F_{\Delta''}(y) = \mathrm{P}(\delta_i'' \leq y) = 1 - \mathrm{P}(\delta_i'' > y) = 1 - \mathrm{P}(\bigcap_j a_{ij} > y) \tag{22}$$

$$= 1 - (1 - F_D(y))^{m+1} \tag{23}$$

$$\tag{24}$$

For convenience, we denote $1 - (1 - F_D(y))^{m+1}$ as $F_\delta(y)$. Next, we derive the distribution ($F_{Z'}(z)$ and $F_{Z''}(z)$ ) of $Z'$ and $Z''$, respectively:

$$F_{Z'}(z) = P(Z' \leq z) = P(\max_{m+1 < i \leq n} \delta_i \leq z) = P(\bigcap_{m+1 < i \leq n} \delta_i \leq z) \tag{25}$$

$$= \begin{cases} F_\Delta(z)^{(n-m+1)} & , z < \mathrm{E}[d_H(X, X^{(m)})] \\ 1 & , else \end{cases} \tag{26}$$

Analogously,

$$F_{Z''}(z) = P(Z'' \leq z) = P(\max_{m+1 < i \leq n} \delta_i \leq z) = P(\bigcap_{m+1 < i \leq n} \delta_i \leq z) \tag{27}$$

$$= \begin{cases} F_\Delta(z)^{(n-m+1)} & , z < \mathrm{E}[d_H(X, X^{(m)})] \\ F_\Delta(z) & , else \end{cases} \tag{28}$$

Thus, we have:

$$\mathrm{E}_{Z' \sim F_{Z'}}[Z'] = \int_0^{+\infty} (1 - F_{Z'}(z)) \, \mathrm{d}z - \int_{-\infty}^0 F_{Z'}(z) \, \mathrm{d}z \tag{29}$$

$$= \int_0^{+\infty} (1 - F_{Z'}(z)) \, \mathrm{d}z \tag{30}$$

$$= \int_0^{\mathrm{E}[d_H(X,X^{(m)})]} (1 - F_{Z'}(z)) \, \mathrm{d}z + \int_{\mathrm{E}[d_H(X,X^{(m)})]}^{+\infty} (1 - F_{Z'}(z)) \, \mathrm{d}z \tag{31}$$

$$= \int_0^{\mathrm{E}[d_H(X,X^{(m)})]} (1 - F_\Delta(z)^{n-m}) \, \mathrm{d}z + \int_{\mathrm{E}[d_H(X,X^{(m)})]}^{+\infty} (1 - 1) \, \mathrm{d}z \tag{32}$$

$$= \int_0^{\mathrm{E}[d_H(X,X^{(m)})]} (1 - F_\Delta(z)^{n-m}) \, \mathrm{d}z \tag{33}$$

and:

$$\mathrm{E}_{Z'' \sim F_{Z''}}[Z''] = \int_{0}^{+\infty} (1 - F_{Z''}(z))\,\mathrm{d}z \tag{34}$$

$$= \int_{0}^{\mathrm{E}[d_H(X,X^{(m)}]} (1 - F_{Z''}(z))\,\mathrm{d}z + \int_{\mathrm{E}[d_H(X,X^{(m)}]}^{+\infty} (1 - F_{Z''}(z))\,\mathrm{d}z \tag{35}$$

$$= \int_{0}^{\mathrm{E}[d_H(X,X^{(m)}]} (1 - F_\Delta(z)^{n-m-1})\,\mathrm{d}z + \int_{\mathrm{E}[d_H(X,X^{(m)}]}^{+\infty} (1 - F_\Delta(z))\,\mathrm{d}z \tag{36}$$

$$\tag{37}$$

Finally,

$$\mathrm{E}_{Z' \sim F_{Z'}}[Z'] - \mathrm{E}_{Z'' \sim F_{Z''}}[Z''] = \int_{\mathrm{E}[d_H(X,X^{(m)}]}^{+\infty} (F_\Delta(z) - 1)\,\mathrm{d}z \leq 0 \tag{38}$$

$$=> \mathrm{E}_{Z' \sim F_{Z'}}[Z'] \leq \mathrm{E}_{Z'' \sim F_{Z''}}[Z''] \tag{39}$$

$$=> \mathrm{E}[\mathrm{d_H}(X, X_T^{(m+1)})] \leq \mathrm{E}[\mathrm{d_H}(X, X_R^{(m+1)})] \tag{40}$$

$$\square$$

## C  PERSISTENT HOMOLOGY

Here, we further provide several explanations on the definition of simplex, simplicial complex, abstract simplicial complex and Vietoris-Rips complex. (a) Simplex: In geometry, a simplex is a generalization of the notion of a triangle or tetrahedron to arbitrary dimensions. The simplex is so-named because it represents the simplest possible polytope made with line segments in any given dimension. For example, a 0-simplex is a point, a 1-simplex is a line segment, and a 2-simplex is a triangle. (b) Simplicial Complex: In topology, it is common to "glue together" simplices to form a simplicial complex. A simplicial complex is a set composed of points, line segments, triangles, and their n-dimensional counterparts. The strict definition of a simplicial complex is that A simplicial complex $K$ is a set of simplices that satisfies the following conditions: 1) Every face of a simplex from $K$ is also in $K$; 2) The non-empty intersection of any two simplices $\sigma_1, \sigma_2 \in K$ is a face of both $\sigma_1$ and $\sigma_2$. (c) Abstract Simplicial Complex The purely combinatorial counterpart to a simplicial complex is an abstract simplicial complex. (d) Vietoris–Rips complex: In topology, the Vietoris–Rips complex, also called the Vietoris complex or Rips complex, is a way of forming a topological space from distances in a set of points. It is an abstract simplicial complex that can be defined from any metric space M and distance $\delta$ by forming a simplex for every finite set of points that has a diameter at most $\delta$. That is, it is a family of finite subsets of M, in which we think of a subset of k points as forming a $(k1)$-dimensional simplex (an edge for two points, a triangle for three points, a tetrahedron for four points, etc.); if a finite set S has the property that the distance between every pair of points in S is at most $\delta$, then we include S as a simplex in the complex. As illustrated in Moor et al. (2020), we can compute the persistent homology of a set of data points $X$ based on this background information.

## D  COMPUTATION PROCEDURE OF TOPOLOGY-PRESERVING LOSS

Here, we further introduce how to retrieve the value of 0-dimensional / 1-dimensional / 2-dimensional persistence diagram from distance matrix with indices provided by the persistence pairings, namely $\mathcal{D}_0^{X_v^{(m)}} \simeq \mathbf{A}^{X_v^{(m)}}[\pi_0^{X_v^{(m)}}]$. In essence, this retrieving procedure equals to how to select retreival indices from 0-dimensional / 1-dimensional / 2-dimensional persistence pairings. Concretely, for 0-dimensional topological features , we select the "destroyer" simplices in the 0-dimensional persistence pairings. For 1-dimensional topological features and 1-dimensional topological features , we regard the maximum edge of the "destroyer simplices" in corresponding persistence pairings as retrieval indices.

# E EXPERIMENTS.

## E.1 ABLATION STUDY

Based on the CADA-VAE Schonfeld et al. (2019), we conduct ablative experiments on CUB, SUN, and AWA2 datasets to verify the effectiveness of our proposed Topology-guided Sampling Strategy and Topology Alignment Module. We first clarify the notations in Tab. 2. TAD denotes our Topology Alignment Module. $TAD_0$ / $TAD_{0-1}$ represents our Topology Alignment Module with preserving 0-dimensional/ 0-dimensional and 1-dimensional topological features. We can see the 4th row with TAD performs a better result than the 2nd row with $TAD_0$ and the 3rd row with $TAD_{0-1}$, indicating the effectiveness of multi-dimensional (especially high dimensional) structure alignment. Then, with the addition of TGSS, the performance is further enhanced, demonstrating the TGSS can achieve better structure alignment. This experiment result is highly compatible with our provided theoretical analysis on TGSS.

Table 2: Ablation studies of TGSS and TAD on CUB, SUN, and AWA2 datasets.

| Method | CUB CZSL acc | CUB GZSL U | CUB GZSL S | CUB GZSL H | SUN CZSL acc | SUN GZSL U | SUN GZSL S | SUN GZSL H | AWA2 CZSL acc | AWA2 GZSL U | AWA2 GZSL S | AWA2 GZSL H |
|---|---|---|---|---|---|---|---|---|---|---|---|---|
| CADA-VAE | 57.9 | 51.6 | 53.5 | 52.4 | 61.6 | 47.2 | 35.7 | 40.6 | 62.6 | 51.6 | 53.5 | 52.4 |
| CADA-VAE + $TAD_0$ | 59.2 | 51.3 | 58.8 | 54.8 | 61.8 | 48.3 | 37.2 | 42.0 | 67.4 | 55.5 | 71.3 | 63.6 |
| CADA-VAE + $TAD_{0-1}$ | 60.3 | 52.7 | 59.2 | 55.8 | 61.9 | 48.5 | 38.1 | 42.7 | 68.2 | 56.2 | 77.3 | 65.1 |
| CADA-VAE + TAD | 62.2 | 53.7 | 58.7 | 56.1 | 62.5 | 48.8 | 39.2 | 43.5 | 68.8 | 58.6 | 76.9 | 66.6 |
| TopoZero (CADA-VAE + TAD + TGSS) | **64.3** | **54.9** | **59.9** | **57.3** | **64.7** | **49.4** | **40.9** | **44.7** | **70.6** | **59.1** | **80.0** | **68.0** |

## E.2 ANALYSIS

**The effectiveness of TAM.** To verify the effectiveness of single Topology Alignment Module, we disentangle it from our overall TopoZero framework. As reported in Tab. 3, although a single TAM can achieve great performance, there exactly exists a distinct performance gap compared with recent sota methods Chen et al. (2021c;b). This is why we introduce an off-the-shelf distribution alignment module into our TopoZero framework.

Table 3: The effectiveness of single TAM.

| Method | CUB | SUN | AWA |
|---|---|---|---|
| TAM | 58.4 | 60.7 | 64.1 |
| TAM + TGSS | **60.5** | **63.1** | **66.4** |

**Compatibility with sota ZSL frameworks.** To verify the compatibility between our TAM branch with sota ZSL framework, we implement our proposed TGSS and TAD on the sota open-source method TransZero Chen et al. (2022a). The CZSL results are listed in Tab. 4. After applying the proposed TGSS and TAM, the performance of TransZero increases to 77.6%, 67.2% and 72.6% on CUB, SUN and AWA datasets respectively. This improvement indicates the effectiveness of the proposed method on the sota attribute-based ZSL framework. Note that the superiority of TransZero benefits from it utilizes semantic attribute vectors of each attribute learned by GloVe Pennington et al. (2014) to improve semantic representation. Through this extra knowledge, recent attribute-based ZSL methods Chen et al. (2021a; 2022b) perform better than others without extra knowledge. Thus, we only verify the compatibility between our TopoZero and Transzero rather than comparing performance directly.

Table 4: Compatibility with sota ZSL framework TransZero.

| Method | CUB | SUN | AWA |
|---|---|---|---|
| TransZero | 76.8 | 65.6 | 70.1 |
| TransZero + Ours | **77.6** | **67.2** | **72.6** |

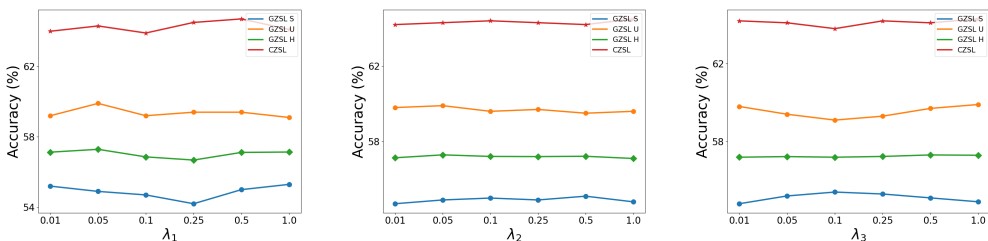

Figure 3: The coarse effects of $\lambda_1$, $\lambda_2$ and $\lambda_3$ on the CUB dataset.

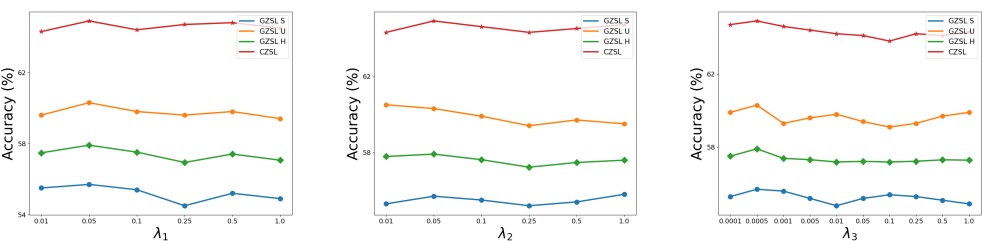

Figure 4: The fine effects of $\lambda_1$, $\lambda_2$ and $\lambda_3$ on the CUB dataset.

**Model Complexity Analysis.** Our TopoZero has a clear intuition of leveraging parallel structure and distribution for advancing ZSL. Such design thus inevitably leads to the first 5 terms in Eq. 18 for multi-dimensional structure alignment and the last 5 terms in Eq. 18 for distribution alignment. Although TopoZero has 4 autoencoders in total, the entire training process is simultaneous and loss weights of all terms in Eq. 18 are the same for all datasets. The consistently significant results on all datasets show that our model is robust and easy to train. Additionally, several losses are formulated with similar forms, which are cooperated for easy optimization, *i.e.* $\mathcal{L}^x_{AE}$ and $\mathcal{L}^a_{AE}$, $\mathcal{L}^x_{CA}$ and $\mathcal{L}^a_{CA}$, $\mathcal{L}^x_{TP}$ and $\mathcal{L}^a_{TP}$. Finally, TAM and DAM are parallel and such disentangle design can make the learning curve smooth and maximize the role of each branch, respectively. Benefiting from this disentangled design, our TopoZero is easy to train compared to HSVA, where the latter adopts coupled framework.

**Hyper-parameter Analysis.** In this part, we further verify the sensitivity of hyper-parameter in our TopoZero by conducting experiments on the CUB dataset, including $\lambda_1$, $\lambda_2$, and $\lambda_3$. As shown in Fig. 3, the performance of TopoZero is of great robustness when varying hyper-parameter from $\{0.01, 0.05, 0.1, 0.25, 0.5, 1.0\}$. Finally, $\lambda_1$, $\lambda_2$ and $\lambda_3$ are set 0.05, 0.05, and 1 in this paper for the better result.

Although this hyper-parameter configuration achieves a great performance on 3 ZSL benchmark datasets, it also raises an interesting question: given these 3 hyper-parameters play distinct role in our TopoZero framework, why their effects are so consistent? For instance, the green lines in Fig. 3 almost present a consistent trending. The reason for this question is that the configuration in the hyper-parameter selection setting is unreasonable, where the candidate range of $\lambda_3$ is small. This hides the role of each term in the objective function since the value of 4-th term (controlled by $\lambda_3$) is far larger than that of 2-nd (controlled by $\lambda_1$) and 3rd (controlled by $\lambda_2$) terms, where the value of $\mathcal{L}^a_{VAE}$ and $\mathcal{L}^a_{VAE}$ in 4-th term is extraordinarily large. Thus, to conduct a detailed hyper-parameter analysis, we extend the range of $\lambda_3$ into $\{0.0001, 0.0005, 0.001, 0.0050.01, 0.05, 0.1, 0.25, 0.5, 1.0\}$. Based on this revision, the individual effects of the three hyper-parameters are expanded remarkably, that is illustrated in Fig. 4. Simultaneously, our TopoZero achieves a higher CZSL accuracy of 64.9% on the cub dataset via this step. In our opinion, this improvement benefits from this more reasonable hyper-parameter selection procedure, which is conducive to getting rid of "hyper-parameter overfitting" via mining the role of each item accurately. Considering this step involves some tricks of hyper-parameter tuning, we only discuss this situation rather than adopting this hyper-parameter configuration for better results.

**Visualization Result.** As shown in Fig. 1 (a) - (c), we utilize persistent homology to visualize the multi-dimensional topological features of TopoZero and HSVA latent structure space. We can

see that our TopoZero topological latent space presents an almost consistent trend in terms of input topological space while HSVA fails, indicating that our TopoZero can preserve more geometry information than HSVA when handling with 'curse of dimensionality' Wang & Chen (2017).

