# OpenReview forum: "TopoZero: Digging into  Topology Alignment on Zero-Shot Learning"
_ICLR.cc/2023/Conference — Submitted to ICLR 2023_

### Official Review · Reviewer_MFWd · 2022-10-23

**Confidence:** 5
**Clarity, Quality, Novelty And Reproducibility:** 1. The framework is clear and some fo…
**Correctness:** 3
**Technical Novelty And Significance:** 2
**Empirical Novelty And Significance:** 2
**Recommendation:** 3

**Strength And Weaknesses:**

Strength:
1. This paper is well-organized and well-written, which is easy to read.
2. The idea of topology alignment is novel, but it is only reflected in sampling data, where the relations among data are not modeled, which is a bit trivial.

Weakness:
1. Figure 1 is difficult to understand. What are the lines in figure 1 (a-c) represent? What does ‘0-dim’ represent?
2. The framework is too completed. It contains 4 autoencoders, which seems an ensemble of the proposed approach and CADA-VAE. There are too many losses in the framework. Is it efficient to learn?
3. The representations in the formulation are inconsistent.
(a)Is ‘x’ in Eq 2 ‘x_max’?
(b)Are ‘Z_v’ in Eq 9 and ‘z_x’ in Eq 11 the same? ‘z_x’ is not mentioned in the paper.
4. All the compared methods are before 2022 and the performance is not good compared with the recent 2022 papers.
5. The proposed method is an incremental work based on HSVA, why not use the same baseline with HSVA. It is difficult to judge whether the improvement is made by the proposed TAD or an ensemble of CADA-VAE.
6. No qualitative analysis is made to show why preserving the global structure is good. What does it benefit? Is it good to generate more effective features? However, no feature visualization is conducted.

**Summary Of The Paper:**

This paper proposes a TopoZero framework to improve structure alignment for common space learning methods. Based on two weaknesses of HSVA (trains model with local structure and loses some high-dimensional structure information), the authors propose a Topology-guided sampling strategy and a Topology Alignment Module. Extensive experiments show the effectiveness of the proposed approach.

**Summary Of The Review:**

The proposed approach aims to preserve the global topology of training samples to train the models, which seems novel. However,  the framework is too complicated, which ensembles CADA-VAE. It is difficult to judge which part improves the performance. Moreover, the compared methods are not recent state-of-the-arts and there is no qualitative analysis of the proposed approach.

---

> ### Author Response · Authors · 2022-11-10
> **Author's Response to Reviewer MFWd**
>
> We thank the reviewer for the constructive comments and for taking the time to thoroughly read this paper. We are glad that you find our work novel and you think that the paper is written with good clarity. Below, we address your comments on weaknesses point-by-point.
>
> **Q1**: What are the lines in figure 1 (a-c) represent? What does ‘0-dim’ represent?
>
> **A1** The blue and red lines represent the topological structure of global data points and batch data points, respectively. 0/1/2-dimensional topological features represent connected components/circles/voids.
>
> **Q2**: The complexity of our method. Is it efficient to learn?
>
> **A2**:
> (i) Complexity. In our opinion, this complexity derives from the distribution alignment module. Our original framework only contains TAM and TGSS and the results are listed below. Although a single TAM can achieve great performance, there exactly exists a distinct performance gap compared with recent sota methods. This is why we introduce an off-the-shelf distribution alignment module into our TopoZero framework， which makes the overall framework complex.
>
> | Method | CUB | SUN | AWA |
> | :----: | :----: |:----: |:----: |
> | TAM | 58.4 | 60.7 | 64.1 |
> | TAM + TGSS | 60.5 | 63.1 | 66.4 |
>
> (ii) Although too many losses exist in our objective function, the learning procedure is efficient. The reason is that the TAM and DAM are parallel and such disentangle design can make the learning curve smooth and maximize the role of each branch, respectively. Benefiting from this disentangled design and our high-dimensional structure preservation ability, our TopoZero outperforms HSVA by a significant improvement.
>
> **Q3**: The representations in the formulation are inconsistent.
>
> **A3**: Thank you for the comment.
> (i) Is ‘$x$’ in Eq 2 ‘$x_max$’? Yes, line 5 in Algorithm 1 illustrates this.
> (ii) Are ‘$Z_v$’ in Eq 9 and ‘$z_x$’ in Eq 11 the same? Yes, we will fix it in the revision. Note that although $z_x$ defined in the 2nd paragraph of Section 3.2, we will polish this part to make the paper reader-friendly.
>
> **Q4**: All the compared methods are before 2022 and the performance is not good compared with the recent 2022 papers.
>
> **A4**: Thanks for the suggestion. In the below table, we add several comparisons with recent/advanced ZSL methods and report the corresponding CZSL result, including VGSE [1], MSDN [2], DOE-ZSL [3]. The symbol “–” indicates no available result. These results show the superior results of TopoZero compared to recent/advanced ZSL methods. Note that the superiority of MSDN benefits from it utilizes semantic attribute vectors of each attribute learned by GloVe [4] to improve semantic representation. Through this extra knowledge, recent attribute-based ZSL methods [5,6] perform better than others without extra knowledge.
> Thus, to make a fair comparison, attribute-based ZSL methods should not be considered in this work.
>
>
> | Method | CUB | SUN | AWA |
> | :----: | :----: |:----: |:----: |
> | VGSE | 56.8 | 41.1 | 66.7 |
> | MSDN | 76.1 | 65.8 | 70.1 |
> | DOE-ZSL | - |  - | 66.4 |
> | Ours | 64.3 | 64.7 | 70.6|
>
> **Q5**: The proposed method is an incremental work based on HSVA, why not use the same baseline with HSVA?
>
> **A5**: Thanks for your advice.
> The reasons behind such design are that (i) The open-source training code of HSVA is not available; (ii) The main contribution of this work is TAD and TGSS.
>
> **Q6**: It is difficult to judge whether the improvement is made by the proposed TAD or an ensemble of CADA-VAE.
>
> **A6**: Good question. During the rebuttal phase, we provide the result of a single TAD with TGSS in A2. The performance of our TAD with TGSS is superior to that of HSVA, indicating the superiority of our TAD and TGSS.
>
> **Q7** No qualitative analysis is made to show why preserving the global structure is good.
>
> **A7**  It's a critical point. We present 2 reasons for this. (i) Structure
>  alignment between visual and semantic domains is important in ZSL due to the heterogeneous nature of the feature representations in the two domains. (ii) A vast body of dimensionality reduction works [1,2,3] reveals that after dimensionality reduction  the  structure is difficult to maintain. Thus, preserving the global structure is conducive to the subsequent structure alignment step.
>
>
> [1] Mokbel, Bassam, et al. "Visualizing the quality of dimensionality reduction." Neurocomputing 112 (2013): 109-123.
>
> [2] Liu, Shusen, et al. "Visualizing high-dimensional data: Advances in the past decade." IEEE transactions on visualization and computer graphics 23.3 (2016): 1249-1268.
>
> [3] Bibal, Adrien, and Benoıt Frénay. "Measuring quality and interpretability of dimensionality reduction visualizations." Safe Machine Learning Workshop at ICLR. 2019.

---

> > ### Comment · Reviewer_MFWd · 2022-11-11
> > **The authors has addressed some of my concerns, but some key problems still exist.**
> >
> > 1. It is still unclear how Figure1 is drawn. How is the topology obtained?  Is the blue line composed of data points?
> > 2. The framework is complicated. TAM has used two autoencoders, but it does not achieve state-of-art performance. If it is combined with CADA-VAE, four autoencoders are used, which is too complicated to train. Moreover, it still not achieves state-of-art performance.
> > 3. The performance is not good compared with the recent state-of-the-art. Although the authors have compared some 2022 approaches in the response, only the ZSL performance is shown. The GZSL performance is still not good. The authors have also performed the TransZero baseline, but the total framework is changed, which is not the approach described in this paper. Moreover,  it is not clear how TransZero is combined with their approach.

---

> > > ### Author Response · Authors · 2022-11-11
> > > **Author's Response to Reviewer MFWd**
> > >
> > > We thank the reviwer for the quick response! We are very happy to help to address your concerns!
> > >
> > > **Q1**: It is still unclear how Figure1 is drawn. How is the topology obtained? Is the blue line composed of data points?
> > >
> > > **A1**: Figure 1 is drawn by an open-source tool, Ripser \url{https://github.com/Ripser/ripser}, which can compute persistent homology up to high dimension.
> > > Given a set of data points, ripser can compute the corresponding persistent homology, which consists of persistence diagrams
> > > {\pi_1, \pi_2, ...} and  persistence pairs {D_1, D_2, ...}.
> > > The $d$-dimensional persistence diagram
> > > $\pi_d$  contains coordinates with the
> > > form $(a, b)$, where $a$ refers to a threshold $\epsilon$ at which
> > > a $d$-dimensional topological feature appears and $b$ refers to a threshold $\epsilon'$ at which it disappears.
> > > The d-dimensional persistence pairs contain indices $(i, j)$ corresponding to simplices $s_i, s_j \in V_{\epsilon}(X)$,
> > > which create and destroy the corresponding topological features determined by $(a, b) \in D_d$.
> > > Thus based on the obtained persistence diagrams and persistence pairs, we can compute the number of alive 0/1/2-dimensional topological features under different threshold $\epsilon$. As such, Figure1 can be drawn and the blue line represents the trend of the number of alive topological features under different threshold $\epsilon$.
> > >
> > >
> > > **Q2**:The framework is complicated. TAM has used two autoencoders, but it does not achieve state-of-art performance. If it is combined with CADA-VAE, four autoencoders are used, which is too complicated to train. Moreover, it still not achieves state-of-art performance.
> > >
> > > **A2**: Our TopoZero has a clear intuition of leveraging parallel structure and distribution for advancing ZSL. Such design thus inevitably leads to the first 5 terms in Eq. 18 for multi-dimensional structure alignment and the last 5  terms in Eq. 18 for distribution alignment.
> > > Although TopoZero has 4 autoencoders in total, the entire training process is simultaneous and loss weights of all terms in Eq. 18 are the same for all datasets. The consistently significant results on all datasets show that our model is robust and easy to train.  Additionally, several losses are formulated with similar forms, which are cooperated for easy optimization, $i.e.$  $L_{AE}^{x}$ and $L_{AE}^{a}$, $L_{CA}^{x}$ and $L_{CA}^{a}$, $L_{TP}^{x}$ and $L_{TP}^{a}$.
> > > Finally, TAM and DAM are parallel and such disentangle design can make the learning curve smooth and maximize the role of each branch, respectively. Benefiting from this disentangled design, our TopoZero is easy to train compared to HSVA, where the latter adopts coupled framework.
> > >
> > > **Q3**: The performance is not good compared with the recent state-of-the-art. Although the authors have compared some 2022 approaches in the response, only the ZSL performance is shown. The GZSL performance is still not good.
> > >
> > > **A3**: Thanks for your suggestion. Here, we add GZSL results in the below tables. These results show the superior results of TopoZero compared to recent/advanced ZSL methods [1,2,3] on both SUN and AWA datasets. While for the CUB dataset, MSDN adopts semantic attribute vectors of each attribute learned by GloVe [4] to improve semantic representation, thus a better result is obtained. However, this is not a fair comparison since the extra representation is introduced.
> > >
> > > - GZSL Results on CUB dataset
> > > | Method | U | S | H |
> > > | :----: | :----: |:----: |:----: |
> > > | VGSE | 24.1 |  45.7 | 31.5 |
> > > | DOE-ZSL |  - | - | - |
> > > | MSDN | 68.7 | 67.5 | 68.1 |
> > > | Ours | 54.9 | 59.9 | 57.3|
> > >
> > > - GZSL Results on SUN dataset
> > > | Method | U | S | H |
> > > | :----: | :----: |:----: |:----: |
> > > | VGSE | 25.5 | 35.7 | 29.8|
> > > | DOE-ZSL |  - | - |  - |
> > > | MSDN | 52.2 | 34.2 | 41.3 |
> > > | Ours | 49.4 | 40.9 | 44.7|
> > >
> > > - GZSL Results on AWA dataset
> > > | Method | U | S | H |
> > > | :----: | :----: |:----: |:----: |
> > > | VGSE | 45.7 |  66.7 | 54.2|
> > > | DOE-ZSL | - | - | 57.6|
> > > | MSDN | 62.0 | 74.5 | 67.7 |
> > > | Ours | 59.1 | 80.0 | 68.0 |
> > >
> > > **Q4**: The authors have also performed the TransZero baseline, but the total framework is changed, which is not the approach described in this paper. Moreover, it is not clear how TransZero is combined with their approach.
> > >
> > > **A4**： The reason of why we perform the TransZero baseline is to demonstrate the plug-and-play property of our TopoZero, as mentioned by Reviewer ugcD. The process of combining our TAM with TransZero is  that our single TAM can obtain a common structure latent representation, which can diectly serve as an another latent representation, thus it can be integrated into any ZSL framework easily.

---

> > > > ### Author Response · Authors · 2022-11-16
> > > > **Update Manuscript**
> > > >
> > > > We have revised our manuscript to include the above details, and have highlighted the additions in the main paper and Appendix with blue color to increase their visibility.  We really appreciate your positive comments and thoughtful suggestions.  We look forward to discussing the new results and would be pleased to answer further questions.

---

> > ### Author Response · Authors · 2022-11-22
> > **Further Discussions**
> >
> > Dear Reviewer MFWd,
> >
> > Thank you for the  encouraging feedback. We are still looking forward to your reply. Would you mind checking our response and confirming if there are unclear explanations?
> >
> > Best, Authors

---

### Official Review · Reviewer_ugcD · 2022-10-23

**Confidence:** 4
**Correctness:** 4
**Technical Novelty And Significance:** 3
**Empirical Novelty And Significance:** 3
**Recommendation:** 6

**Clarity, Quality, Novelty And Reproducibility:**

The proposed methods are clearly presented and the paper is easy to follow; There is some novelty in the batch sampling strategy and topology preserving strategies for ZSL; The experimental results should be reproducible given the details of experimental settings in the paper.


**Strength And Weaknesses:**

++ The proposed methods facilitate the idea of topology preserving during representation learning which has been proven important for zero-shot learning. The proposed TGSS has been theoretically and empirically validated in this work.

++ The proposed method is justified theoretically and empirically. The experiments are thorough and convincing.

-- The authors fail to compare with more recent/advanced ZSL approaches in the experiments.

-- I wonder if the TGSS strategy also works for other ZSL frameworks than CADA-VAE.

-- The authors should have discussed the impact of batch sizes on the performance of TGSS. Why is the batch size set as 50 when the number of classes could be as large as 150/645? Is there any simple baseline methods as alternatives to TGSS? e.g., class-balanced sampling for each batch?

**Summary Of The Paper:**

The paper proposesa Topology-guided Sampling Strategy (TGSS) to mitigate the distribution gap between sampled and global data points for Zero-Shot Learning. In addition, a Topology Alignment Module (TAM) is proposed to perserve multi-dimensional geometry structure in latent visual and semantic space. The proposed method is evaluated on several benchmark datasets for ZSL and generalised ZSL and achieves superior performance.

**Summary Of The Review:**

The authors aim to design an approach to facilitating topology preserving in ZSL from a novel perspective. The proposed method is proved to be effective when combined with a classic ZSL framework CADA-VAE which, however, is a relatively old technique for ZSL. The authors fail to compare with more advanced ZSL approaches.

---

> ### Author Response · Authors · 2022-11-10
> **Author's Response to Reviewer ugcD**
>
> We thank the reviewer for the constructive comments. We are glad that you believe that our  paper is easy to follow and novel. We have carefully gone through your comments and addressed each of them point-by-point.
>
> **Q1**: The authors fail to compare with more recent/advanced ZSL approaches in the experiments.
>
> **A1**: Thanks for the suggestion. In the below table, we add several comparisons with recent/advanced ZSL methods and report the corresponding CZSL results, including VGSE [1], MSDN [2], and DOE-ZSL [3]. The symbol “–” indicates no available result. These results show the superior results of TopoZero compared to recent/advanced ZSL methods. Note that the superiority of MSDN benefits from that it utilizes semantic attribute vectors of each attribute learned by GloVe [4] to improve semantic representation. Through this extra knowledge, recent attribute-based ZSL methods [5,6] perform better than others without extra knowledge.
> Thus, to make a fair comparison, attribute-based ZSL methods should not be considered in this work.
>
>
> | Method | CUB | SUN | AWA |
> | :----: | :----: |:----: |:----: |
> | VGSE | 56.8 | 41.1 | 66.7 |
> | MSDN | 76.1 | 65.8 | 70.1 |
> | DOE-ZSL | - |  - | 66.4 |
> | Ours | 64.3 | 64.7 | 70.6|
>
>
> **Q2**: I wonder if the TGSS strategy also works for other ZSL frameworks than CADA-VAE.
>
> **A2**: It's a great advice. We implement our proposed TGSS and TAD on the sota open-source method TransZero [5] and spent the past day conducting the experiments. The CZSL results are listed below. After applying the proposed TGSS and TAD, the performance of TransZero increases to 77.6\%, 67.2\%, and 72.6\% on CUB, SUN and AWA, respectively. This improvement indicates the effectiveness of the proposed method on the attribute-based ZSL framework, which is also the recent sota ZSL method. It is worth mentioning that we started running the experiments immediately after reading your feedback, and there was not enough time for us to tune the parameters in TransZero + Ours, but used the default values instead.
>
> | Method | CUB | SUN | AWA |
> | :----: | :----: |:----: |:----: |
> | TransZero | 76.8 | 65.6 | 70.1 |
> | TransZero + Ours | 77.6 | 67.2 | 72.6|
>
>
> **Q3**:  Why is the batch size set as 50 when the number of classes could be as large as 150/645?
>
> **A3** To make a fair comparison, the batch size are set consistent with that in CADA-VAE [7], which is our baseline method.
>
> **Q4**: Is there any simple baseline methods as alternatives to TGSS? e.g., class-balanced sampling for each batch?
>
> **A4**: Good insight. Thanks to this constructive suggestion, we find an another property of our TGSS, namely plug-and-play. In our opinion, class-balanced sampling (CBS) strategy can be compatible with our TGSS since the latter can improve the local geometry structure of sampled data points. To verity this compatibility, we conduct experiments by adding TBS on our TopoZero and TopoZero (w/o TGSS). The results in below table show the effectiveness of TBS and the compatibility between our TGSS and CBS.
>
> | Method | CUB | SUN | AWA |
> | :---- | :----: |:----: |:----: |
> | TopoZero w/o TGSS | 62.2 | 62.5 | 68.6 |
> | TopoZero w/o TGSS + CBS | 63.5 | 63.4 | 69.8 |
> | TopoZero | 64.3 | 64.7 | 70.6 |
> | TransZero + CBS | 64.8 | 64.9 | 71.3|
>
>
> ** We are pleased that your appreciation of our paper. we hope you feel the flaws are addressed in our response. **
>
>
> [1] Xu, Wenjia, et al. "VGSE: Visually-Grounded Semantic Embeddings for Zero-Shot Learning." Proceedings of the IEEE/CVF Conference on Computer Vision and Pattern Recognition. 2022.
>
> [2] Chen, Shiming, et al. "MSDN: Mutually Semantic Distillation Network for Zero-Shot Learning." Proceedings of the IEEE/CVF Conference on Computer Vision and Pattern Recognition. 2022.
>
> [3] Geng, Yuxia, et al. "Disentangled ontology embedding for zero-shot learning." Proceedings of the 28th ACM SIGKDD Conference on Knowledge Discovery and Data Mining. 2022.
>
> [4] Pennington, Jeffrey, Richard Socher, and Christopher D. Manning. "Glove: Global vectors for word representation." Proceedings of the 2014 conference on empirical methods in natural language processing (EMNLP). 2014.
>
> [5] Chen, Shiming, et al. "Transzero: Attribute-guided transformer for zero-shot learning." AAAI. Vol. 2. 2022.
>
> [6] Chen, Shiming, et al. "Transzero++: Cross Attribute-Guided Transformer for Zero-Shot Learning." 	arXiv:2112.08643.
>
> [7] Schonfeld, Edgar, et al. "Generalized zero-and few-shot learning via aligned variational autoencoders." Proceedings of the IEEE/CVF Conference on Computer Vision and Pattern Recognition. 2019.

---

### Official Review · Reviewer_FV59 · 2022-10-23

**Confidence:** 5
**Correctness:** 3
**Technical Novelty And Significance:** 4
**Empirical Novelty And Significance:** 3
**Recommendation:** 8

**Clarity, Quality, Novelty And Reproducibility:**

The paper is presented at very high quality. Intuitive illustrations are provided which makes the idea very easy to follow. The focus on the exploration of topological properties of visual-semantic information is novel and can have wide impacts in ZSL and ML domains.

**Details Of Ethics Concerns:**

Not applicable.

**Strength And Weaknesses:**

+ The motivation for topology structure alignment is clear and strong.
+ The theoretical analysis of TGSS is solid and inspiring.
+ The paper is well presented and easy to follow.
- The overall Objective Function (Eq 18 and 19 should be labelled once only) contains duplicates. For example, the reconstruction is considered multiple times in AE, TP and CA.
- The hyperparameter evaluation is odd. Given AE TP and CA are counting, visual-semantic reconstruction, topologic alignment, and distributional alignment, why the effects of the three hyperparameters are so consistent? It usually needs to find a balance between complementary objective functions. Otherwise, at least one of the terms must be useless.
- Despite the intuitive rationale in Figure 1 and solid theoretical analysis in 3.1, the idea is lack quantitative analysis supported by empirical examples.

**Summary Of The Paper:**

This paper proposes a topology-guided sampling strategy as a parallel pipeline to CADA-VAE. Empirical and theoretical analysis of the topology property is provided. The method is evaluated on the three common benchmarks of CUB SUN and AWA and achieves promising performance.

**Summary Of The Review:**

Overall, this work meets the standard expected by ICLR. However, the model design contains duplication and redundancy which normally indicates risks in "hyper-parameter overfitting". The evaluation of hyperparameters in the appendix is very contradictory and needs further elaboration.

---

> ### Author Response · Authors · 2022-11-10
> **Author's Response to Reviewer FV59**
>
> We thank the reviewer for the valuable comments. We are glad that you like our work, feel that our paper is presented at very high quality and can have wide impacts in ZSL and ML domains. We also thank you for pointing out the weakness of our hyper-parameter analysis. Below, we address your comments on weaknesses point-by-point.
>
> **Q1**: Given AE TP and CA are counting, visual-semantic reconstruction, topologic alignment, and distributional alignment, why the effects of the three hyperparameters are so consistent?
>
> **A1**: Thanks for this instructive suggestion. The reason for this phenomenon is that our configuration in the hyper-parameter selection setting is unreasonable, where the candidate range of $\lambda_3$ is small. This hides the role of each term in the objective function since the value of CA (controlled by $\lambda_3$) is far larger than that of AE and TP, where the value of $L_{VAE}^{a}$ and ${L}_{VAE}^{x}$ in CA is extraordinarily large. Thus, to conduct a reasonable hyper-parameter analysis, we extend the range of $\lambda_3$ into {0.0001, 0.0005, 0.001, 0.005，0.01, 0.05, 0.1, 0.25, 0.5, 1.0}. Based on this revision, the individual effects of the three hyper-parameters are expanded remarkably which can be reflected clearly during the hyper-parameter analysis phase. Simultaneously, our TopoZero achieves a higher accuracy of 64.9% on the cub dataset via this step. In our opinion, this improvement benefits from this more reasonable hyper-parameter selection procedure, which is conducive to getting rid of "hyper-parameter overfitting" via mining the role of each item accurately.
>
> **Q2**: The idea is lack quantitative analysis supported by empirical examples.
>
> **A2**: Per your suggestion, we add 2 types of experiments:
>
> (i) Compatibility with sota ZSL frameworks. We implement our proposed TGSS and TAD on the sota open-source method TransZero [1] and spent the past day conducting the experiments. The CZSL results are listed below. After applying the proposed TGSS and TAD, the performance of TransZero increases to 77.6\%, 67.2\% and 72.6\% on CUB, SUN and AWA datasets respectively. This improvement indicates the effectiveness of the proposed method on the attribute-based ZSL framework, which is also the recent sota ZSL method. It is worth mentioning that we started running the experiments immediately after reading your feedback, and there was not enough time for us to tune the parameters in TransZero + Ours, but used the default values instead.
>
> | Method | CUB | SUN | AWA |
> | :---- | :----: |:----: |:----: |
> | TransZero | 76.8 | 65.6 | 70.1 |
> | TransZero + Ours | 77.6 | 67.2 | 72.6|
>
> (ii) The plug-and-play property of our TGSS.
> To verify this, we conduct experiments to compare our TGSS and a typical data sampling strategy, i.e., class-balanced sampling (CBS) strategy.
> The results in the below table show the effectiveness of TBS and the compatibility between our TGSS and CBS. Thus, our TGSS can be well compatible with two well-known data sampling strategies, demonstrating its plug-and-play property.
>
> | Method | CUB | SUN | AWA |
> | :---- | :----: |:----: |:----: |
> | TopoZero w/o TGSS | 62.2 | 62.5 | 68.6 |
> | TopoZero w/o TGSS + CBS | 63.5 | 63.4 | 69.8 |
> | TopoZero | 64.3 | 64.7 | 70.6 |
> | TransZero + CBS | 64.8 | 64.9 | 71.3|
>
> **Q3**: The overall Objective Function (Eq 18 and 19 should be labeled once only) contains duplicates.
>
> **A3**: Thanks for your suggestion, we will remove redundant equations in the revision.
>
>
> **We are pleased that your appreciation on our paper. we hope you feel the flaws are addressed in our response.**
>
> [1] Chen, Shiming, et al. "Transzero: Attribute-guided transformer for zero-shot learning." AAAI. Vol. 2. 2022.

---

> > ### Author Response · Authors · 2022-11-16
> > **Author's Response to Reviewer FV59**
> >
> > We have revised our manuscript to include the above details, and have highlighted "A1"/ "A2"  in the Section E.2 with the blue/red color. The revision of "A3"  is placed on the below part of Equ. 18 with red color.  We appreciate the reviewer's recognition of our work and insightful suggestions.

---

### Official Review · Reviewer_ijnG · 2022-10-25

**Confidence:** 4
**Correctness:** 3
**Technical Novelty And Significance:** 2
**Empirical Novelty And Significance:** 2
**Recommendation:** 5

**Clarity, Quality, Novelty And Reproducibility:**

To mitigate the gap between sampled data within a mini-batch and global data, a topology-guided sampling strategy (TGSS) is proposed, which has certain novelties.

On the other hand, the paper is not self-contained, which hinders understanding of the contents of the paper. For example, the explanation of Persistent Homology on page 2 is not familiar to many readers, so the reviewer would like to see an intuitive and qualitative explanation in the text and a detailed explanation in the Appendix. The lack of explanations makes the middle part of the description in 3.2 difficult to understand.

The paper is not well written, and it is difficult to read through. For example, the description in 3.1.1 does not explain more than that the calculations are performed using Equations 1 and 2. Qualitative and intuitive descriptions of each equation should be provided.

The description of equation 2 is particularly insufficient. At the time of reading this text, it would be more appropriate to write $\mathcal{C}$ instead of $\mathcal{U}$, and it would be more helpful to write $x$ as $x_{max}$.

Figure 2 is difficult to understand. In particular, the explanation of (d)-(f) is insufficient.

The evaluation of the proposed method is insufficient. The model is a combination of CADA-VAE and TAP, thus the performance of TAP alone should be presented.

Also, each batch must be sampled according to Algorithm 1. The reviewer would like to know how long this computation takes, and would like to see some experiments on the order of computation and actual computation time.

What is the problem of "The latent visual and semantic space fails to preserve multiple dimensional geometry structure, especially high dimensional structure information." The reviewer would like to see a detailed explanation as to whether this problem has been solved along with the basis for the experiment.

There are some spelling errors.
- In equation 2, the $($ in $(x_0$ is not necessary.
- In equation 2, it is $X$, not $\mathcal{X}$.
- In the definition of $X$ in 3.1.2, it is $x_m$, not $x_n$.


**Strength And Weaknesses:**

- Strength

-- A topology guided sampling strategy (TGSS) is proposed to mitigate the gap between sampled data in a mini-batch and global data.

-- A theoretical analysis is presented for TGSS.

TopoZero shows good performance on standard ZSL benchmark datasets.

- Weaknesses

-- The paper is not self-contained, which hinders understanding of the paper.

-- The paper is poorly written, making it difficult to read the paper smoothly.

-- There are some typos.

-- Evaluations of the proposed method are insufficient.


**Summary Of The Paper:**

This paper proposes a topology-guided sampling strategy (TGSS) to mitigate the gap between sampled data within a mini-batch and global data. The proposed model, which is called TopoZero, consists of a topology alignment module (TAM) and a distribution alignment module. TAM is capable of preserving multidimensional geometric structures in each of the latent visual and semantic spaces. Experimental results show that TopoZero performs well on standard ZSL benchmark datasets.

**Summary Of The Review:**

Although the proposed method, called TGSS, has a certain degree of novelty and shows improvement in performance from the baselines, it is not considered to have reached the stage of publishing at this time due to a lack of experiments and insufficient explanation due to the low quality of the writing.

---

> ### Author Response · Authors · 2022-11-10
> **Author's Response to Reviewer ijnG (Part-1 out of 2)**
>
> We thank the reviewer for the constructive comments and for taking the time to thoroughly read this paper. We are glad that you think our work is novel and some experimental results are great. Below, we address your comments on weaknesses point-by-point.
>
> **Q1**: The reviewer would like to see an intuitive and qualitative explanation in the text and a detailed explanation in the Appendix.
>
> **A1**: Per your suggestion, we add a comprehensive explanation corresponding to Persistent Homology in the revision, that is organized as follows:
>
> (i) Main paper: Persistent homology is a tool for computing topological features, namely connectivity-based features of a data set at different spatial resolutions, e.g., connected components in 0-dimensional, cycles in 1-dimensional, and voids in 2-dimensional topological features. More persistent features can be found over a wide range of spatial scales and  are deemed more likely to represent true features of the underlying space.
>
> We first introduce the concept of simplicial homology. For a simplicial complex $R$, i.e. a generalised graph with higher-order connectivity information such as cliques, simplicial homology employs matrix reduction algorithms to
> assign $R$ a family of groups, namely homology groups. The $d$-th homology group
> $H_{d}(R)$ of
> $R$ contains $d$-dimensional topological features, such as connected components ($d = 0$), cycles/tunnels ($d = 1$), and voids ($d = 2$). Homology groups are typically summarised by their
> ranks, thereby obtaining a simple invariant signature of a manifold.
> For example, a circle in $R^2$ has one feature with $d = 1$ (a cycle), and one feature with $d
> = 0$ (a connected component).
> Moreover, we introduce how to compute a Persistent Homology when given a point cloud $X$.
> Firstly, we denote the Vietoris-Rips complex of $X$ at scale $\epsilon$ as $V_{\epsilon}(X)$.
> Then, we can obtain the Persistent Homology ${PH}(V_{\epsilon}(X))$ of a Vietoris-Rips complex, which consists of persistence diagrams
> {\pi_1, \pi_2, ...} and  persistence pairs {D_1, D_2, ...}.
> The $d$-dimensional persistence diagram
> $\pi_d$  contains coordinates with the
> form $(a, b)$, where $a$ refers to a threshold $\epsilon$ at which
> a $d$-dimensional topological feature appears and $b$ refers to a threshold $\epsilon'$ at which it disappears.
> The d-dimensional persistence pairs contain indices $(i, j)$ corresponding to simplices $s_i, s_j \in V_{\epsilon}(X)$,
> which create and destroy the corresponding topological features determined by $(a, b) \in D_d$.
>
> (ii) Appendix:  Here, we further provide several explanations on the definition of simplex, simplicial complex, abstract simplicial complex and Vietoris-Rips complex. (a) Simplex: In geometry, a simplex is a generalization of the notion of a triangle or tetrahedron to arbitrary dimensions. The simplex is so-named because it represents the simplest possible polytope made with line segments in any given dimension. For example, a 0-simplex is a point, a 1-simplex is a line segment, and a 2-simplex is a triangle. (b) Simplicial Complex: In topology, it is common to "glue together" simplices to form a simplicial complex. A simplicial complex is a set composed of points, line segments, triangles, and their n-dimensional counterparts. The strict definition of a simplicial complex is that A simplicial complex $K$ is a set of simplices that satisfies the following conditions: 1) Every face of a simplex from $K$ is also in $K$; 2) The non-empty intersection of any two simplices $\sigma_1, \sigma_2 \in K$ is a face of both $\sigma_1$ and $\sigma_2$. (c) Abstract Simplicial Complex： The purely combinatorial counterpart to a simplicial complex is an abstract simplicial complex. (d)  Vietoris–Rips complex: In topology, the Vietoris–Rips complex, also called the Vietoris complex or Rips complex, is a way of forming a topological space from distances in a set of points. It is an abstract simplicial complex that can be defined from any metric space M and distance δ by forming a simplex for every finite set of points that has a diameter at most δ. That is, it is a family of finite subsets of M, in which we think of a subset of k points as forming a (k − 1)-dimensional simplex (an edge for two points, a triangle for three points, a tetrahedron for four points, etc.); if a finite set S has the property that the distance between every pair of points in S is at most δ, then we include S as a simplex in the complex.

---

> > ### Author Response · Authors · 2022-11-10
> > **Author's Response to Reviewer ijnG (Part-2 out of 2)**
> >
> > **Q2**: Figure 2 is difficult to understand. In particular, the explanation of (d)-(f) is insufficient.
> >
> > **A2**: Thanks for your suggestion. We add a clear explanation for this. The threshold in Fig. (d)-(f) refers to the $\epsilon$ in $V_{\epsilon}(X)$. The vertical axis of Fig. (d)-(f) represents the number of alive 0-2-dimensional topological features at scale $\epsilon$. Based on these incremental explanations, we can make 2 conclusions from the visualization result of  Fig. (d)-(f): (i) Compared to the input space, HSVA latent space can only preserve 0-dimensional
> > topological features, indicating some high dimensional structure representation is lost during the
> > dimension reduction phase. (ii) In contrast, our TopoZero latent space can preserve more accurate topological features by taking advantage of our proposed Topology Alignment Module.
> >
> > **Q3**: The performance of TAM alone should be presented.
> >
> > **A3**:  It's a great advice. We conducted this experiment before we submit the manuscript and the results are listed below. Although a single TAM can achieve great performance, there exactly exists a distinct performance gap compared with recent sota methods. This is why we introduce an off-the-shelf distribution alignment module into our TopoZero framework.
> >
> > | Method | CUB | SUN | AWA |
> > | :----: | :----: |:----: |:----: |
> > | TAM | 58.4 | 60.7 | 64.1 |
> > | TAM + TGSS | 60.5 | 63.1 | 66.4 |
> >
> > **Q4**: The reviewer would like  to see some experiments on the order of computation and actual computation time.
> >
> > **A4**：The order of computation is described as follows: (i) Compute the distance matrix $A$ between $X$ and $X$. (2) The computation process of all the following steps can be easily obtained via the index of $X_{b2}$ and $A$. Thus, the computation time of our proposed TGSS can be ignored.
> >
> > **Q5**: What is the problem of "The latent visual and semantic space fails to preserve multiple dimensional geometry structure, especially high dimensional structure information."
> >
> > **A5**: This problem refers to that after dimensionality reduction from input space to latent space, the 0-dimensional topological structure can maintain while the 1/2-dimensional topological structures fail. The statistical result is presented in the orange and blue lines in Fig. (d)-(f). Besides, the negligible gap between the orange and green line in Fig. (d)-(f) represents that our Topozero can preserve high dimensional structure information.
> >
> > **Q6**: typos and writing.
> >
> > **A6** Thanks for your suggestion. We will correct all the mentioned typos and misunderstanding parts in the revision. Furthermore, by introducing a clear explanation of the background knowledge on persistent homology, we make the paper more reader-friendly.
> >
> > **We are pleased that you find our idea novel. we hope you feel the flaws are addressed in our response.**

---

> > > ### Author Response · Authors · 2022-11-15
> > > **Author's Response to Reviewer ijnG**
> > >
> > > We have revised our manuscript to include the above details, and have highlighted the additions with blue color.
> > > Furthermore, we thank the reviewer for his/her valuable feedback which will help a lot to make our work more reader-friendly.
> > > Finally, we have tried our best to address all the concerns and provided explanations to all questions. If there are still unclear parts to you, please kindly let us know. We are very glad to further discuss them.

---

> > > > ### Comment · Reviewer_ijnG · 2022-12-02
> > > > **Thank you for your answers**
> > > >
> > > > Dear authors,
> > > >
> > > > Thank you for your explanations.
> > > >
> > > > Regards,
> > > >
> > > > Rev. ijnG

---

> > > > > ### Author Response · Authors · 2022-12-02
> > > > > **Author's Response to Reviewer ijnG**
> > > > >
> > > > > Dear reviewer ijnG,
> > > > >
> > > > > Thanks again for your valuable efforts in reviewing. We hope you feel the flaws are addressed in our response. We would be happy to answer any further queries you might have before the end of the discussion period.
> > > > >
> > > > > Best, Authors

---

> ### Author Response · Authors · 2022-11-22
> **Further Discussions**
>
> Dear Reviewer ijnG,
>
> Thanks again for your valuable efforts in reviewing. We are still looking forward to your reply. Would you mind checking our response and confirming if there are unclear explanations?
>
> Best,
> Authors

---

### Author Response · Authors · 2022-11-17
**Author-Reviewer Discussion Due Approaching**

Dear reviewers,

Thanks again for your valuable comments. We kindly remind reviewers that the discussion period is coming to an end and we would like to know if there are any remaining changes or questions you would like us to answer.
We would be happy to answer any further queries you might have before the end of the discussion period. Do let us know if you found our response satisfactory or/and wish to take forward the discussion. Thanks!

Regards,

The Authors

---

### Decision · Program_Chairs · 2023-01-20

**Decision:**

Reject

**Justification For Why Not Higher Score:**

This paper lacks clear motivations and presentation and its technical part is the combination of the existing losses. More details can be found in my meta review. In sum, the current version of this paper is not mature.

**Justification For Why Not Lower Score:**

N/A

**Metareview: Summary, Strengths And Weaknesses:**

Based on the collected information from all reviewers and my personal judgment, I can make the recommendation on this paper, **reject**. Here are the comments that I summarized, which include my opinion and evidence.

**Research Problem**

This paper studies the well-defined generalized zero-shot learning problem

**Presentation**

In general, this paper is difficult to follow. Many format issues including the abstract format on the OpenReview system, missing punctuation, incorrect citation format, spelling errors, and typos. Some sentences and paragraphs lack logic. For example, the authors criticize the existing design as sub-optimal. How to verify this point? What is the optimum? Can the proposed method achieve the optimum? In the first paragraph on the second page, the authors illustrate the two issues in HSVA. However, in what follows, the authors use the whole paragraph to introduce persistent homology. Figure 1 is also confusing, where many notations and concepts are used without any definition.

**Motivation**

The motivations come from two aspects. (1) The structure in a mini-batch is different from the global one and (2) the latent visual and semantic space fail to preserve multiple dimensional geometry structure. Frankly speaking, these two motivations are too general and not strong.

**Philosophy**

This paper lacks philosophy on how to tackle the above challenges. In Section 3, the authors directly introduce the algorithm, without any rationality.

**Technique**

Several reviewers have concerns about too complicated techniques, which contain many losses and hyperparameters.  Please see their detailed comments.

**Experiments**

The authors only provide the performance on the benchmark datasets. Unfortunately, the authors fail to verify their motivations.

No objection was raised from the reviewers on the rejection recommendation.

**Summary Of Ac-Reviewer Meeting:**

This is not a borderline paper.